# Efficient Regret Minimization Algorithm for Extensive-Form Correlated Equilibrium[*]

**Gabriele Farina**
Computer Science Department
Carnegie Mellon University
gfarina@cs.cmu.edu

**Chun Kai Ling**
Computer Science Department
Carnegie Mellon University
chunkail@cs.cmu.edu

**Fei Fang**
Institute for Software Research
Carnegie Mellon University
feif@cs.cmu.edu

**Tuomas Sandholm**
Computer Science Department, CMU
Strategic Machine, Inc.
Strategy Robot, Inc.
Optimized Markets, Inc.
sandholm@cs.cmu.edu

## Abstract

Self-play methods based on regret minimization have become the state of the art for computing Nash equilibria in large two-players zero-sum extensive-form games. These methods fundamentally rely on the hierarchical structure of the players' sequential strategy spaces to construct a regret minimizer that recursively minimizes regret at each decision point in the game tree. In this paper, we introduce the first efficient regret minimization algorithm for computing extensive-form *correlated* equilibria in large two-player *general-sum* games with no chance moves. Designing such an algorithm is significantly more challenging than designing one for the Nash equilibrium counterpart, as the constraints that define the space of correlation plans lack the hierarchical structure and might even form cycles. We show that some of the constraints are redundant and can be excluded from consideration, and present an efficient algorithm that generates the space of extensive-form correlation plans incrementally from the remaining constraints. This structural decomposition is achieved via a special convexity-preserving operation that we coin *scaled extension*. We show that a regret minimizer can be designed for a scaled extension of any two convex sets, and that from the decomposition we then obtain a global regret minimizer. Our algorithm produces feasible iterates. Experiments show that it significantly outperforms prior approaches and for larger problems it is the only viable option.

## 1 Introduction

In recent years, self-play methods based on regret minimization, such as counterfactual regret minimization (CFR) [Zinkevich *et al.*, 2007] and its faster variants [Tammelin *et al.*, 2015; Brown *et al.*, 2017; Brown and Sandholm, 2019a] have emerged as powerful tools for computing Nash equilibria in large extensive-form games, and have been instrumental in several recent milestones in poker [Bowling *et al.*, 2015; Brown and Sandholm, 2017a,b; Moravčík *et al.*, 2017; Brown and Sandholm, 2019b]. These methods exploit the hierarchical structure of the sequential strategy spaces of the players to construct a regret minimizer that recursively minimizes regret locally at each decision

---

[*]The full version of this paper is available on arXiv.

point in the game tree. This has inspired regret-based algorithms for other solution concepts in game theory, such as extensive-form perfect equilibria [Farina *et al.*, 2017], Nash equilibrium with strategy constraints [Farina *et al.*, 2017, 2019a,b; Davis *et al.*, 2019], and quantal-response equilibrium [Farina *et al.*, 2019a].

In this paper, we give the first efficient regret-based algorithm for finding an *extensive-form correlated equilibrium (EFCE)* [von Stengel and Forges, 2008] in two-player general-sum games with no chance moves. EFCE is a natural extension of the correlated equilibrium (CE) solution concept to the setting of extensive-form games. Here, the strategic interaction of rational players is complemented by a *mediator* that privately recommends behavior, but does not *enforce it*: it is up to the mediator to make recommendations that the players are incentivized to follow. Designing a regret minimization algorithm that can efficiently search over the space of extensive-form correlated strategies (known as *correlation plans*) is significantly more difficult than designing one for Nash equilibrium. This is because the constraints that define the space of correlation plans lack the hierarchical structure of sequential strategy spaces and might even form cycles. Existing general-purpose regret minimization algorithms, such as follow-the-regularized-leader [Shalev-Shwartz and Singer, 2007] and mirror descent, as well as those proposed by Gordon *et al.* [2008] in the context of convex games, are not practical: they require the evaluation of proximal operators (generalized projections problems) or the minimization of linear functions on the space of extensive-form correlation plans. In the former case, no distance-generating function is known that can be minimized efficiently over this space, while in the latter case current linear programming technology does not scale to large games, as we show in the experimental section of this paper. The regret minimization algorithm we present in this paper computes the next iterate in *linear* time in the dimension of the space of correlation plans.

We show that some of the constraints that define the polytope of correlation plans are redundant and can be eliminated, and present an efficient algorithm that generates the space of correlation plans incrementally from the remaining constraints. This structural decomposition is achieved via a special convexity-preserving operation that we coin *scaled extension*. We show that a regret minimizer can be designed for a scaled extension of any two convex sets, and that from the decomposition we then obtain a global regret minimizer. Experiments show that our algorithm significantly outperforms prior approaches—the LP-based approach [von Stengel and Forges, 2008] and a very recent subgradient descent algorithm [Farina *et al.*, 2019c]—and for larger problems it is the only viable option.

## 2 Preliminaries

Extensive-form games (EFGs) are played on a game tree. Each node in the game tree belongs to a player, who acts at that node; for the purpose of this paper, we focus on two-player games only. Edges leaving a node correspond to actions that can be taken at that node. In order to capture private information, the game tree is supplemented with *information sets*. Each node belongs to exactly one information set, and each information set is a nonempty set of tree nodes for the same Player $i$, which are the set of nodes that Player $i$ cannot distinguish among, given what they have observed so far. We will focus on *perfect-recall* EFGs, that is, EFGs where no player forgets what the player knew earlier. We denote by $\mathcal{I}_1$ and $\mathcal{I}_2$ the sets of all information sets that belong to Player 1 and 2, respectively. All nodes that belong to an information set $I \in \mathcal{I}_1 \cup \mathcal{I}_2$ share the same set of available actions (otherwise the player acting at those nodes would be able to distinguish among them); we denote by $A_I$ the set of actions available at information set $I$. We define the set of *sequences* of Player $i$ as the set $\Sigma_i := \{(I, a) : I \in \mathcal{I}_i, a \in A_I\} \cup \{\varnothing\}$, where the special element $\varnothing$ is called *empty sequence*. Given an information set $I \in \mathcal{I}_i$, we denote by $\sigma(I)$ the *parent sequence* of $I$, defined as the last pair $(I', a') \in \Sigma_i$ encountered on the path from the root to any node $v \in I$; if no such pair exists (that is, Player $i$ never acts before any node $v \in I$), we let $\sigma(I) = \varnothing$. We (recursively) define a sequence $\tau \in \Sigma_i$ to be a *descendent* of sequence $\tau' \in \Sigma_i$, denoted by $\tau \succeq \tau'$, if $\tau = \tau'$ or if $\tau = (I, a)$ and $\sigma(I) \succeq \tau'$. We use the notation $\tau \succ \tau'$ to mean $\tau \succeq \tau' \wedge \tau \neq \tau'$. Figure 1 shows a small example EFG; black round nodes belong to Player 1, white round nodes belong to Player 2, action names are not shown, gray round sets define information sets, and the numbers along the edges define concise names for sequences (for example, '7' denotes sequence $(\text{D}, a)$ where $a$ is the leftmost action at D).

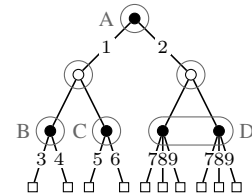

Figure 1: Small example.

**Sequence-Form Strategies** In the sequence-form representation [Romanovskii, 1962; Koller *et al.*, 1996; von Stengel, 1996], a strategy for Player $i$ is compactly represented via a vector $\boldsymbol{x}$ indexed

by sequences $\sigma \in \Sigma_i$. When $\sigma = (I, a)$, the entry $x[\sigma] \geq 0$ defines the product of the probabilities according to which Player $i$ takes their actions on the path from the root to information set $I$, up to and including action $a$; furthermore, $x[\varnothing] = 1$. Hence, in order to be a valid sequence-form strategy, $x$ must satisfy the 'probability mass conservation' constraint: for all $I \in \mathcal{I}_i$, $\sum_{a \in A_I} x[(I, a)] = x[\sigma(I)]$. That is, every information sets partitions the probability mass received from the parent sequence onto its actions. In this sense, the constraints that define the space of sequence-form strategies naturally exhibit a hierarchical structure.

## 2.1 Extensive-Form Correlated Equilibria

*Extensive-form correlated equilibrium (EFCE)* [von Stengel and Forges, 2008] is a natural extension of the solution concept of *correlated equilibrium (CE)* [Aumann, 1974] to extensive-form games. In EFCE, a mediator privately reveals recommendations to the players as the game progresses. These recommendations are *incremental*, in the sense that recommendations for the move to play at each decision point of the game are revealed only if and when the decision point is reached. This is in contrast with CE, where recommendations *for the whole game* are privately revealed upfront when the game starts. Players are free to not follow the recommended moves, but once a player does not follow a recommendation, he will not receive further recommendations. In an EFCE, the recommendations are incentive-compatible—that is, the players are motivated to follow all recommendations. EFCE and CE are good candidates to model strategic interactions in which intermediate forms of centralized control can be achieved [Ashlagi *et al.*, 2008].

In a recent preprint, Farina *et al.* [2019c] show that in two-player perfect-recall extensive-form games, an EFCE that guarantees a social welfare (that is, sum of player's utilities) at least $\tau$ is the solution to a bilinear saddle-point problem, that is an optimization problem of the form $\min_{\boldsymbol{x} \in \mathcal{X}} \max_{\boldsymbol{y} \in \mathcal{Y}} \boldsymbol{x}^\top \boldsymbol{A} \boldsymbol{y}$, where $\mathcal{X}$ and $\mathcal{Y}$ are convex and compact sets and $\boldsymbol{A}$ is a matrix of real numbers. In the case of EFCE, $\mathcal{X} = \Xi$ is known as the *polytope of correlation plans* (see Section 2.2) and $\mathcal{Y}$ is the convex hull of certain sequence-form strategy spaces. In general, $\Xi$ cannot be captured by a polynomially small set of constraints, since computing an optimal EFCE in a two-player perfect-recall game is computationally hard [von Stengel and Forges, 2008].[2] However, in the special case of games with no chance moves, this is not the case, and $\Xi$ is the intersection of a polynomial (in the game tree size) number of constraints, as discussed in the next subsection. In fact, most of the current paper is devoted to studying the structure of $\Xi$. We will largely ignore $\mathcal{Y}$, for which an efficient regret minimizer can already be built, for instance by using the theory of *regret circuits* [Farina *et al.*, 2019b] (see also Appendix A in the full paper). Similarly, we will not use any property of matrix $\boldsymbol{A}$ (except that it can be computed and stored efficiently).

## 2.2 Polytope of Extensive-Form Correlation Plans in Games with no Chance Moves

In their seminal paper, von Stengel and Forges [2008] characterize the constraints that define the space of extensive-form correlation plans $\Xi$ in the case of two-player perfect-recall games *with no chance moves*. The characterization makes use of the following two concepts:

**Definition 1** (Connected information sets, $I_1 \rightleftharpoons I_2$)**.** *Let $I_1, I_2$ be information sets for Player 1 and 2, respectively. We say that $I_1$ and $I_2$ are* connected, *denoted $I_1 \rightleftharpoons I_2$, if there exist two nodes $u \in I_1, v \in I_2$ such that $u$ is on the path from the root to $v$, or $v$ is on the path from the root to $u$.*

**Definition 2** (Relevant sequence pair, $\sigma_1 \bowtie \sigma_2$)**.** *Let $\sigma_1 \in \Sigma_1, \sigma_2 \in \Sigma_2$. We say that $(\sigma_1, \sigma_2)$ is a* relevant sequence pair, *and write $\sigma_1 \bowtie \sigma_2$, if either $\sigma_1$ or $\sigma_2$ or both is the empty sequence, or if $\sigma_1 = (I_1, a_1)$ and $\sigma_2 = (I_2, a_2)$ and $I_1 \rightleftharpoons I_2$. Similarly, given $\sigma_1 \in \Sigma_1$ and $I_2 \in \mathcal{I}_2$, we say that $(\sigma_1, I_2)$ forms a relevant sequence-information set pair, and write $\sigma_1 \bowtie I_2$, if $\sigma_1 = \varnothing$ or if $\sigma_1 = (I_1, a_1)$ and $I_1 \rightleftharpoons I_2$ (a symmetric statement holds for $I_1 \bowtie \sigma_2$).*

**Definition 3** (von Stengel and Forges [2008])**.** *In a two-player perfect-recall extensive-form game with no chance moves, the space $\Xi$ of* correlation plans *is a convex polytope containing* nonnegative *vectors indexed over relevant sequences pairs, and is defined as*

$$\Xi := \left\{ \boldsymbol{\xi} \geq \boldsymbol{0} : \begin{array}{ll} \bullet\ \xi[\varnothing, \varnothing] = 1 & \\ \bullet\ \sum_{a \in A_I} \xi[(I_1, a), \sigma_2] = \xi[\sigma(I_1), \sigma_2] & \forall I_1 \in \mathcal{I}_1, \sigma_2 \in \Sigma_2\ s.t.\ I_1 \bowtie \sigma_2 \\ \bullet\ \sum_{a \in A_J} \xi[\sigma_1, (I_2, a)] = \xi[\sigma_1, \sigma(I_2)] & \forall I_2 \in \mathcal{I}_2, \sigma_1 \in \Sigma_1\ s.t.\ \sigma_1 \bowtie I_2 \end{array} \right\}.$$

*In particular, $\Xi$ is the intersection of at most $1 + |\mathcal{I}_1| \cdot |\Sigma_2| + |\Sigma_1| \cdot |\mathcal{I}_2|$ constraints, a polynomial number in the input game size.*

## 2.3 Regret Minimization and Relationship with Bilinear Saddle-Point Problems

A regret minimizer is a device that supports two operations: (i) RECOMMEND, which provides the next decision $x^{t+1} \in \mathcal{X}$, where $\mathcal{X}$ is a nonempty, convex, and compact subset of a Euclidean space $\mathbb{R}^n$; and (ii) OBSERVELOSS, which receives/observes a convex loss function $\ell^t$ that is used to evaluate decision $x^t$ [Zinkevich, 2003]. In this paper, we will consider linear loss functions, which we represent in the form of a vector $\ell^t \in \mathbb{R}^n$. A regret minimizer is an *online* decision maker in the sense that each decision is made by taking into account only past decisions and their corresponding losses. The quality metric for the regret minimizer is its *cumulative regret* $R^T$, defined as the difference between the loss cumulated by the sequence of decisions $x^1, \ldots, x^T$ and the loss that would have been cumulated by the *best-in-hindsight time-independent* decision $\hat{x}$. Formally, $R^T := \sum_{t=1}^T \langle \ell^t, x^t \rangle - \min_{\hat{x} \in \mathcal{X}} \sum_{t=1}^T \langle \ell^t, \hat{x} \rangle$. A 'good' regret minimizer has $R^T$ sublinear in $T$; this property is known as *Hannan consistency*. Hannan consistent regret minimizers can be used to converge to a solution of a *bilinear saddle-point problem* (Section 2.1). To do so, two regret minimizers, one for $\mathcal{X}$ and one for $\mathcal{Y}$, are set up so that at each time $t$ they observe loss vectors $\ell_x^t := -Ay^t$ and $\ell_y^t := A^\top x^t$, respectively, where $x^t \in \mathcal{X}$ and $y^t \in \mathcal{Y}$ are the decisions output by the two regret minimizers. A well-known folk theorem asserts that in doing so, at time $T$ the average decisions $(\bar{x}^T, \bar{y}^T) := (\frac{1}{T} \sum_{t=1}^T x^t, \frac{1}{T} \sum_{t=1}^T y^t)$ have *saddle-point gap* (a standard measure of how close a point is to being a saddle-point) $\gamma(\bar{x}^T, \bar{y}^T) := \max_{\hat{x} \in \mathcal{X}} \hat{x}^\top A \bar{y}^T - \min_{\hat{y} \in \mathcal{Y}} (\bar{x}^T)^\top A \hat{y}$ bounded above by $\gamma(\bar{x}^T, \bar{y}^T) \leq (R_{\mathcal{X}}^T + R_{\mathcal{Y}}^T)/T$ where $R_{\mathcal{X}}^T$ and $R_{\mathcal{Y}}^T$ are the cumulative regrets of the regret minimizers. Since the regrets grow sublinearly, $\gamma(\bar{x}^T, \bar{y}^T) \to 0$ as $T \to +\infty$. As discussed in the introduction, this approach has been extremely successful in computational game theory.

# 3 Scaled Extension: A Convexity-Preserving Operation for Incrementally Constructing Strategy Spaces

In this section, we introduce a new convexity-preserving operation between two sets. We show that it provides an alternative way of constructing the strategy space of a player in an extensive-form game that is different from the construction based on convex hulls and Cartesian products described by Farina *et al.* [2019b]. Our new construction enables one to *incrementally extend* the strategy space in a top-down fashion, whereas the construction by Farina *et al.* [2019b] was bottom-up. Most importantly, as we will show in Section 3.1, this new operation enables one to incrementally, recursively construct the extensive-form correlated strategy space (again in a top-down fashion).

**Definition 4.** *Let $\mathcal{X}$ and $\mathcal{Y}$ be nonempty, compact and convex sets, and let $h : \mathcal{X} \to \mathbb{R}_+$ be a nonnegative affine real function. The* scaled extension *of $\mathcal{X}$ with $\mathcal{Y}$ via $h$ is defined as the set*

$$\mathcal{X} \overset{h}{\triangleleft} \mathcal{Y} := \{(x, y) : x \in \mathcal{X}, \ y \in h(x)\mathcal{Y}\}.$$

Since we will be composing multiple scaled extensions together, it is important to verify that the operation above not only preserves convexity, but also preserves the non-emptiness and compactness of the sets (a proof of the following Lemma is available in Appendix B in the full paper):

**Lemma 1.** *Let $\mathcal{X}, \mathcal{Y}$ and $h$ be as in Definition 4. Then $\mathcal{X} \overset{h}{\triangleleft} \mathcal{Y}$ is nonempty, compact and convex.*

## 3.1 Construction of the Set of Sequence-Form Strategies

The scaled extension operation can be used to construct the polytope of a perfect-recall player's strategy in sequence-form in an extensive-form game. We illustrate the approach in the small example of Figure 1; the generalization to any extensive-form strategy space is immediate. As noted in Section 2, any valid sequence-form strategy must satisfy probability mass constraints, and can be constructed incrementally in a top-down fashion, as follows (in the following we refer to the same naming scheme as in Figure 1 for the sequences of Player 1):

  i. First, the empty sequence is set to value $x[\varnothing] = 1$.
  ii. (Info set A) Next, the value $x[\varnothing]$ is partitioned into the two non-negative values $x[1]+x[2]=x[\varnothing]$.

iii. (Info set B) Next, the value $x[1]$ is partitioned into two non-negative values $x[3] + x[4] = x[1]$.
iv. (Info set C) Next, the value $x[1]$ is partitioned into two non-negative values $x[5] + x[6] = x[1]$.
v. (Info set D) Next, the value $x[2]$ is partitioned into 3 non-negative values $x[7]+x[8]+x[9]=x[2]$.

The incremental choices in the above recipe can be directly translated—in the same order—into set operations by using scaled extensions, as follows:

i. First, the set of all feasible values of sequence $x[\varnothing]$ is the singleton $\mathcal{X}_0 := \{1\}$.
ii. Then, the set of all feasible values of $(x[\varnothing], x[1], x[2])$ is the set $\mathcal{X}_1 := \mathcal{X}_0 \times \Delta^2 = \mathcal{X}_0 \lhd^{h_1} \Delta^2$, where $h_1$ is the linear function $h_1 : \mathcal{X}_0 \ni x[\varnothing] \mapsto x[\varnothing]$ (the identity function).
iii. In order to characterize the set of all feasible values of $(x[\varnothing], \ldots, x[4])$ we start from $\mathcal{X}_1$, and *extend* any element $(x[\varnothing], x[1], x[2]) \in \mathcal{X}_1$ with the two sequences $x[3]$ and $x[4]$, drawn from the set $\{(x[3], x[4]) \in \mathbb{R}_2^+ : x[3] + x[4] = x[1]\} = x[1]\Delta^2$. We can express this extension using scaled extension: $\mathcal{X}_2 := \mathcal{X}_1 \lhd^{h_2} \Delta^2$, where $h_2 : \mathcal{X}_1 \ni (x[\varnothing], x[1], x[2]) \mapsto x[1]$.
iv. Similarly, we can extend every element in $\mathcal{X}_2$ to include $(x[5], x[6]) \in x[1]\Delta^2$: in this case, $\mathcal{X}_3 := \mathcal{X}_2 \lhd^{h_3} \Delta^2$, where $h_3 : \mathcal{X}_2 \ni (x[\varnothing], x[1], x[2], x[3], x[4]) \mapsto x[1]$.
v. The set of all feasible $(x[\varnothing], .., x[9])$ is $\mathcal{X}_4 := \mathcal{X}_3 \lhd^{h_4} \Delta^3$, where $h_4 : \mathcal{X}_3 \ni (x[\varnothing], \ldots, x[6]) \mapsto x[2]$.

Hence, the polytope of sequence-form strategies for Player 1 in Figure 1 can be expressed as $\{1\} \lhd^{h_1} \Delta^2 \lhd^{h_2} \Delta^2 \lhd^{h_3} \Delta^2 \lhd^{h_4} \Delta^3$, where the scaled extension operation is intended as left associative.

### 3.2 Regret Minimizer for Scaled Extension

It is always possible to construct a regret minimizer for $\mathcal{Z} = \mathcal{X} \lhd^h \mathcal{Y}$, where $h(\boldsymbol{x}) = \langle \boldsymbol{a}, \boldsymbol{x} \rangle + b$, starting from a regret minimizer for $\mathcal{X} \subseteq \mathbb{R}^m$ and $\mathcal{Y} \subseteq \mathbb{R}^n$. The fundamental technical insight of the construction is that, given any vector $\boldsymbol{\ell} = (\boldsymbol{\ell}_x, \boldsymbol{\ell}_y) \in \mathbb{R}^m \times \mathbb{R}^n$, the minimization of a linear function $\boldsymbol{z} \mapsto \langle \boldsymbol{\ell}, \boldsymbol{z} \rangle$ over $\mathcal{Z}$ can be split into two separate linear minimization problems over $\mathcal{X}$ and $\mathcal{Y}$:

$$\min_{\boldsymbol{z} \in \mathcal{Z}} \langle \boldsymbol{\ell}, \boldsymbol{z} \rangle = \min_{\boldsymbol{x} \in \mathcal{X}, \boldsymbol{y} \in \mathcal{Y}} \left\{ \langle \boldsymbol{\ell}_x, \boldsymbol{x} \rangle + h(\boldsymbol{x}) \langle \boldsymbol{\ell}_y, \boldsymbol{y} \rangle \right\} = \min_{\boldsymbol{x} \in \mathcal{X}} \left\{ \langle \boldsymbol{\ell}_x, \boldsymbol{x} \rangle + h(\boldsymbol{x}) \min_{\boldsymbol{y} \in \mathcal{Y}} \langle \boldsymbol{\ell}_y, \boldsymbol{y} \rangle \right\}$$
$$= \min_{\boldsymbol{x} \in \mathcal{X}} \left\{ \langle \boldsymbol{\ell}_x + \boldsymbol{a} \cdot \min_{\boldsymbol{y} \in \mathcal{Y}} \langle \boldsymbol{\ell}_y, \boldsymbol{y} \rangle, \boldsymbol{x} \rangle \right\} + b \cdot \min_{\boldsymbol{y} \in \mathcal{Y}} \langle \boldsymbol{\ell}_y, \boldsymbol{y} \rangle.$$

Thus, it is possible to break the problem of minimizing regret over $\mathcal{Z}$ into two regret minimization subproblems over $\mathcal{X}$ and $\mathcal{Y}$ (more details in Appendix C in the full paper). In particular:

**Proposition 1.** *Let $RM_\mathcal{X}$ and $RM_\mathcal{Y}$ be two regret minimizer over $\mathcal{X}$ and $\mathcal{Y}$ respectively, and let $R_\mathcal{X}^T, R_\mathcal{Y}^T$ denote their cumulative regret at time $T$. Then, Algorithm 1 provides a regret minimizer over $\mathcal{Z}$ whose cumulative regret $R_\mathcal{Z}^T$ is bounded above as $R_\mathcal{Z}^T \leq R_\mathcal{X}^T + h^* R_\mathcal{Y}^T$, where $h^* := \max_{\boldsymbol{x} \in \mathcal{X}} h(\boldsymbol{x})$.*

---

**Algorithm 1** Regret minimizer over the scaled extension $\mathcal{X} \lhd^h \mathcal{Y}$.

| | |
|---|---|
| 1: **function** RECOMMEND() | 1: **function** OBSERVELOSS($\boldsymbol{\ell}^t = (\boldsymbol{\ell}_x^t, \boldsymbol{\ell}_y^t)$) |
| 2:    $\boldsymbol{x}^t \leftarrow \text{RM}_\mathcal{X}.\text{RECOMMEND}()$ | 2:    $\boldsymbol{y}^t \leftarrow \text{RM}_\mathcal{Y}.\text{RECOMMEND}()$ |
| 3:    $\boldsymbol{y}^t \leftarrow \text{RM}_\mathcal{Y}.\text{RECOMMEND}()$ | 3:    $\tilde{\boldsymbol{\ell}}_x^t \leftarrow \boldsymbol{\ell}_x^t + \langle \boldsymbol{\ell}_y^t, \boldsymbol{y}^t \rangle \cdot \boldsymbol{a}$ |
| 4:    **return** $(\boldsymbol{x}^t, h(\boldsymbol{x}^t)\boldsymbol{y}^t)$ | 4:    $\text{RM}_\mathcal{X}.\text{OBSERVELOSS}(\tilde{\boldsymbol{\ell}}_x^t)$ |
| | 5:    $\text{RM}_\mathcal{Y}.\text{OBSERVELOSS}(\boldsymbol{\ell}_y^t)$ |

---

Algorithm 1 can be composed recursively to construct a regret minimizer for any set that is expressed via a chain of scaled extensions, such as the polytope of sequence-form strategies (Section 3.1) or that of extensive-form correlation plans (Section 4). When used on the polytope of sequence-form strategies, Algorithm 1 coincides with the CFR algorithm if all regret minimizers for the individual simplexes in the chain of scaled extensions are implemented using the regret matching algorithm [Hart and Mas-Colell, 2000].

## 4 Unrolling the Structure of the Correlated Strategy Polytope

In this section, we study the combinatorial structure of the polytope of correlated strategies (Section 2.2) of a two-player perfect-recall extensive-form game with no chance moves. The central result of this section, Theorem 1, asserts that the correlated strategy polytope $\Xi$ can be expressed via a chain of scaled extensions. This matches the similar result regarding the sequence-form strategy polytope that we discussed in Section 3.1. However, unlike the sequence-form strategy polytope, the

constraints that define the correlated strategy polytope do not exhibit a natural hierarchical structure: the constraints that define $\Xi$ (Definition 3) are such that the same entry of the correlation plan $\boldsymbol{\xi}$ can appear in multiple constraints, and furthermore the constraints will in general form cycles. This makes the problem of unrolling the structure of $\Xi$ significantly more challenging.

The key insight is that some of the constraints that define $\Xi$ are redundant (that is, implied by the remaining constraints) and can therefore be safely eliminated. Our algorithm identifies one such set of redundant constraints, and removes them. The set is chosen in such a way that the remaining constraints can be laid down in a hierarchical way that can be captured via a chain of scaled extensions.

## 4.1 Example

Before we delve into the technical details of the construction, we illustrate the key idea of the algorithm in a small example. In particular, consider the small game tree of Figure 2 (left), where we used the same conventions as in Section 2 and Figure 1. All sequence pairs are relevant; the set of constraints that define $\Xi$ is shown in Figure 2 (middle).

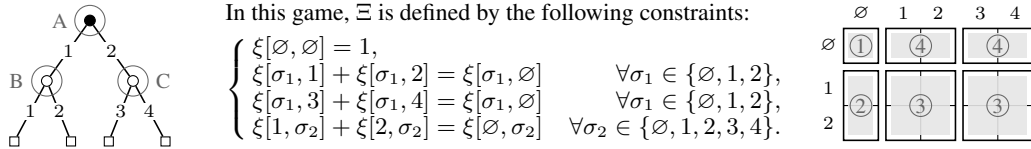

In this game, $\Xi$ is defined by the following constraints:

$$\begin{cases} \xi[\varnothing, \varnothing] = 1, \\ \xi[\sigma_1, 1] + \xi[\sigma_1, 2] = \xi[\sigma_1, \varnothing] & \forall \sigma_1 \in \{\varnothing, 1, 2\}, \\ \xi[\sigma_1, 3] + \xi[\sigma_1, 4] = \xi[\sigma_1, \varnothing] & \forall \sigma_1 \in \{\varnothing, 1, 2\}, \\ \xi[1, \sigma_2] + \xi[2, \sigma_2] = \xi[\varnothing, \sigma_2] & \forall \sigma_2 \in \{\varnothing, 1, 2, 3, 4\}. \end{cases}$$

Figure 2: (Left) Example game (Section 4.1). (Middle) Constraints that define $\Xi$ in the example game. (Right) Fill-in order of $\boldsymbol{\xi}$. The cell at the intersection of row $\sigma_1$ and column $\sigma_2$ represents the entry $\xi[\sigma_1, \sigma_2]$ of $\boldsymbol{\xi}$.

In order to generate all possible correlation plans $\boldsymbol{\xi} \in \Xi$, we proceed as follows. First, we assign $\xi[\varnothing, \varnothing] = 1$. Then, we partition $\xi[\varnothing, \varnothing]$ into two non-negative values $(\xi[1, \varnothing], \xi[2, \varnothing]) \in \xi[\varnothing, \varnothing]\Delta^2$ in accordance with the constraint $\xi[1, \varnothing] + \xi[2, \varnothing] = \xi[\varnothing, \varnothing]$. Next, using the constraints $\xi[\sigma_1, 1] + \xi[\sigma_1, 2] = \xi[\sigma_1, \varnothing]$ and $\xi[\sigma_1, 3] + \xi[\sigma_1, 4] = \xi[\sigma_1, \varnothing]$, we pick values $(\xi[\sigma_1, 1], \xi[\sigma_1, 2]) \in \xi[\sigma_1, \varnothing]\Delta^2$ and $(\xi[\sigma_1, 3], \xi[\sigma_1, 4]) \in \xi[\sigma_1, \varnothing]\Delta^2$ for $\sigma_1 \in \{1, 2\}$. So far, our strategy for filling the correlation plan has been to *split* entries according to the information structure of the players. As shown in Section 3.1, these steps can be expressed via scaled extension operations.

Next, we fill in the four remaining entries in $\xi$, that is $\xi[\varnothing, \sigma_2]$ for $\sigma_2 \in \{1, 2, 3, 4\}$, in accordance with constraint $\xi[1, \sigma_2] + \xi[2, \sigma_2] = \xi[\varnothing, \sigma_2]$. In this step, we are not splitting any value; rather, we fill in $\xi[\varnothing, \sigma_2]$ in the only possible way (that is, $\xi[\varnothing, \sigma_2] = \xi[1, \sigma_2] + \xi[2, \sigma_2]$), by means of a linear combination of already-filled-in entries. This operation can be also expressed via scaled extensions, with the singleton set $\{1\}$: $\{(\xi[1, \sigma_2], \xi[2, \sigma_2], \xi[\varnothing, \sigma_2])\} = \{(\xi[1, \sigma_2], \xi[2, \sigma_2])\} \triangleleft^h \{1\}$, where $h : (\xi[1, \sigma_2], \xi[2, \sigma_2]) \mapsto \xi[1, \sigma_2] + \xi[2, \sigma_2]$ (note that $h$ respects the requirements of Definition 4). This way, we have filled in all entries in $\xi$. However, only 9 out of the 11 constraints have been taken into account in the construction, and we still need to verify that the two leftover constraints $\xi[\varnothing, 1] + \xi[\varnothing, 2] = \xi[\varnothing, \varnothing]$ and $\xi[\varnothing, 3] + \xi[\varnothing, 4] = \xi[\varnothing, \varnothing]$ are automatically satisfied by our way of filling in the entries of $\boldsymbol{\xi}$. Luckily, this is always the case: by construction, $\xi[\varnothing, 1] + \xi[\varnothing, 2] = (\xi[1, 1] + \xi[1, 2]) + (\xi[2, 1] + \xi[2, 2]) = \xi[1, \varnothing] + \xi[2, \varnothing] = \xi[\varnothing, \varnothing]$ (the proof for $\xi[\varnothing, 3] + \xi[\varnothing, 4]$ is analogous). We summarize the construction steps pictorially in Figure 2 (right).

**Remark 1.** *Similar construction that starts from assigning values for $\xi[\varnothing, \sigma_2]$ ($\sigma_2 \in \{1, 2, 3, 4\}$ using constraints $\xi[\varnothing, 1] + \xi[\varnothing, 2] = \xi[\varnothing, \varnothing], \xi[\varnothing, 3] + \xi[\varnothing, 4] = \xi[\varnothing, \varnothing]$ and fills out $\xi[\sigma_1, \sigma_2]$ for $(\sigma_1, \sigma_2) \in \{1, 2\} \times \{1, 2, 3, 4\}$ would have not been successful: if $(\xi[1, 1], \xi[1, 2])$ and $(\xi[1, 3], \xi[1, 4])$ are filled in independently, there is no way of guaranteeing that $\xi[1, 1] + \xi[1, 2] = \xi[1, 3] + \xi[1, 4]$ ($= \xi[1, \varnothing]$) as required by the constraints.*

## 4.2 An Unfavorable Case that Cannot Happen in Games with No Chance Moves

We now show that there exist game instances in which the general approach used in the previous subsection fails. In particular, consider a relevant sequence pair $(\sigma_1, \sigma_2)$ such that both $\sigma_1$ and $\sigma_2$ are parent sequences of two information sets of Player 1 and Player 2 respectively, and assume that all sequence pairs in the game are relevant. Then, no matter what the order of operations is, the situation described in Remark 1 cannot be avoided. Luckily, in two-player perfect-recall games with no chance moves, one can prove that this occurrence never happens (see Appendix D in the full paper for a proof):

**Proposition 2.** *Consider a two-player perfect-recall game with no chance moves, and let $(\sigma_1, \sigma_2)$ be a relevant sequence pair, let $I_1, I_1'$ be two distinct information sets of Player 1 such that $\sigma(I_1) = \sigma(I_1') = \sigma_1$, and let $I_2, I_2'$ be two distinct information sets of Player 2 such that $\sigma(I_2) = \sigma(I_2') = \sigma_2$. It is not possible that both $I_1 \rightleftharpoons I_2$ and $I_1' \rightleftharpoons I_2'$.*

In other words, if $I_1 \rightleftharpoons I_2$, then any pair of sequences $(\sigma_1', \sigma_2')$ where $\sigma_1'$ belongs to $I_1'$ and $\sigma_2'$ belongs to $I_2'$ is *irrelevant*. As we show in the next subsection, this is enough to yield a polynomial-time algorithm to 'unroll' the process of filling in the entries of $\boldsymbol{\xi} \in \Xi$ in any two-player perfect-recall extensive-form game with no chance moves. The following definition is crucial for that algorithm:

**Definition 5.** *Let $(\sigma_1, \sigma_2)$ be a relevant sequence pair, and let $I_1 \in \mathcal{I}_1$ be an information set for Player 1 such that $\sigma(I_1) = \sigma_1$. Information set $I_1$ is called* critical *for $\sigma_2$ if there exists at least one $I_2 \in \mathcal{I}_2$ with $\sigma(I_2) = \sigma_2$ such that $I_1 \rightleftharpoons I_2$. (A symmetric definition holds for an $I_2 \in \mathcal{I}_2$.)*

It is a simple corollary of Proposition 2 that for any relevant sequence pair, *at least one* player has *at most one* critical information set for the opponent's sequence. We call such a player *critical* for that relevant sequence pair.

## 4.3 A Polynomial-Time Algorithm that Decomposes $\Xi$ using Scaled Extensions

In this section, we present the central result of the paper: an efficient algorithm that expresses $\Xi$ as a chain of scaled extensions of simpler sets. In particular, as we have already seen in Section 4.1, each set in the decomposition is either a simplex (when *splitting* an already-filled-in entry) or the singleton set $\{1\}$ (when *summing* already filled-in entries and assigning the result to a new entry of $\boldsymbol{\xi}$).

The algorithm consists of a recursive function, DECOMPOSE, which takes three arguments: a relevant sequence pair $(\sigma_1, \sigma_2)$, a subset $\mathcal{S}$ of the set of all relevant sequence pairs, and a set $\mathcal{D}$ of vectors with entries indexed by the elements in $\mathcal{S}$. $\mathcal{S}$ represents the set of indices of $\boldsymbol{\xi}$ that have already been filled in, while $\mathcal{D}$ is the set of all partially-filled-in correlation plans (see Section 4.1). The decomposition for the whole polytope $\Xi$ is obtained by evaluating DECOMPOSE$((\varnothing, \varnothing), \mathcal{S} = \{(\varnothing, \varnothing)\}, \mathcal{D} = \{(1)\})$, which corresponds to the starting situation in which only the entry $\xi[\varnothing, \varnothing]$ has been filled in (with the value 1 as per Definition 3). Each call to DECOMPOSE returns a pair $(\mathcal{S}', \mathcal{D}')$ of updated indices and partial vectors, to reflect the new entries that were filled in during the call. Each call to DECOMPOSE$((\sigma_1, \sigma_2), \mathcal{S}, \mathcal{D})$ works as follows:

- First, the algorithm finds one critical player for the relevant sequence pair $(\sigma_1, \sigma_2)$ (see end of Section 4.2). Assume without loss of generality that Player 1 is critical (the other case is symmetric), and let $\mathcal{I}^* \subseteq \mathcal{I}_1$ be the set of critical information sets for $\sigma_2$ that belong to Player 1. By definition of critical player, $\mathcal{I}^*$ is either a singleton or it is an empty set.
- For each $I \in \mathcal{I}_1$ such that $\sigma(I) = \sigma_1$ and $I \bowtie \sigma_2$, we:
  - Fill in all entries $\{\xi[(I^*, a), \sigma_2] : a \in A_I\}$ by splitting $\xi[\sigma_1, \sigma_2]$. This is reflected by updating the set of filled-in-indices $\mathcal{S} \leftarrow \mathcal{S} \cup \{((I, a), \sigma_2)\}$ and extending $\mathcal{D}$ via a scaled extension: $\mathcal{D} \leftarrow \mathcal{D} \triangleleft^h \Delta^{|A_I|}$ where $h$ extracts $\xi[\sigma_1, \sigma_2]$ from any partially-filled-in vector.
  - Then, for each $a \in A_I$ we assign $(\mathcal{S}, \mathcal{D}) \leftarrow$ DECOMPOSE$(((I, a), \sigma_2), \mathcal{S}, \mathcal{D})$.

  After this step, all the indices in $\{(\sigma_1', \sigma_2') : \sigma_1' \succ \sigma_1, \sigma_2' \succeq \sigma_2\} \cup \{(\sigma_1, \sigma_2)\}$ have been filled in, and none of the indices in $\{(\sigma_1, \sigma_2') : \sigma_2' \succ \sigma_2\}$ have been filled in yet.
- Finally, we fill out all indices in $\{(\sigma_1, \sigma_2') : \sigma_2' \succ \sigma_2\}$. We do so by iterating over all information sets $J \in \mathcal{I}_2$ such that $\sigma(J) \succeq \sigma_2$ and $\sigma_1 \bowtie J$. For each such $J$, we split into two cases, according to whether $\mathcal{I}^* = \{I^*\}$ (for some $I^* \in \mathcal{I}_1$, as opposed to $\mathcal{I}^*$ being empty) and $J \rightleftharpoons I^*$, or not:
  - If $\mathcal{I}^* = \{I^*\}$ and $J \rightleftharpoons I^*$, then for all $a \in A_J$ we fill in the sequence pair $\xi[\sigma_1, (J, a)]$ by assigning its value in accordance with the constraint $\xi[\sigma_1, (J, a)] = \sum_{a^* \in A_{I^*}} \xi[(I^*, a^*), (J, a)]$ via the scaled extension $\mathcal{D} \leftarrow \mathcal{D} \triangleleft^h \{1\}$ where the linear function $h$ maps a partially-filled-in vector to the value of $\sum_{a^* \in A_{I^*}} \xi[(I^*, a^*), (J, a)]$.
  - Otherwise, we fill in the entries $\{\xi[\sigma_1, (J, a)] : a \in A_J\}$, by splitting the value $\xi[\sigma_1, \sigma(J)]$. In other words, we let $\mathcal{D} \leftarrow \mathcal{D} \triangleleft^h \Delta^{|A_J|}$ where $h$ extracts the entry $\xi[\sigma_1, \sigma(J)]$ from a partially-filled-in vector in $\mathcal{D}$.
- At this point, all the entries corresponding to indices $\tilde{\mathcal{S}} = \{(\sigma_1', \sigma_2') : \sigma_1' \succeq \sigma_1, \sigma_2' \succeq \sigma_2\}$ have been filled in, and we return $(\mathcal{S} \cup \tilde{\mathcal{S}}, \mathcal{D})$.

Every call to DECOMPOSE increases the cardinality of $\mathcal{S}$ by at least one unit. Since $\mathcal{S}$ is a subset of the set of relevant sequence pairs, and since the total number of relevant sequence pair is polynomial

in the input game tree size, the algorithm runs in polynomial time. See Appendix E in the full paper for pseudocode, as well as a proof of correctness of the algorithm. Since every change to $\mathcal{D}$ is done via scaled extensions (with either a simplex or the singleton set $\{1\}$), we conclude that:

**Theorem 1.** *In a two-player perfect-recall EFG with no chance moves, the space of correlation plans $\Xi$ can be expressed via a sequence of scaled extensions with simplexes and singleton sets:*

$$\Xi = \{1\} \overset{h_1}{\lhd} \mathcal{X}_1 \overset{h_2}{\lhd} \mathcal{X}_2 \overset{h_3}{\lhd} \cdots \overset{h_n}{\lhd} \mathcal{X}_n, \text{ where, for } i = 1, \ldots, n, \text{ either } \mathcal{X}_i = \Delta^{s_i} \text{ or } \mathcal{X}_i = \{1\}, \quad (1)$$

*and $h_i(\cdot) = \langle \boldsymbol{a_i}, \cdot \rangle$ is a linear function. Furthermore, an exact algorithm exists to compute such expression in polynomial time.*

We can recursively use Algorithm 1 on the expression (1) to obtain a regret minimizer for $\Xi$. The resulting algorithm, shown in Algorithm 3 of Appendix F in the full paper, is contingent on a choice of "local" regret minimizers $\mathrm{RM}_i$ for each of the simplex domains $\Delta^{s_i}$ in (1). By virtue of Algorithm 1, if each local regret minimizer $\mathrm{RM}_i$ for $\Delta^{s_i}$ runs in linear time (i.e., computes recommendations and observes losses by running an algorithm whose complexity is linear in $s_i$)[3], then the overall regret minimization algorithm for $\Xi$ runs in linear time in the number of relevant sequence pairs of the game. Furthermore, Proposition 1 immediately implies that if each $\mathrm{RM}_i$ is Hannan consistent, then so is our overall algorithm for $\Xi$. Putting these observations together, we conclude:

**Theorem 2.** *For any two-player extensive-form game with no chance moves, there exists a Hannan consistent regret minimizer for $\Xi$ that runs in linear time in the number of relevant sequence pairs.*

## 5   Experimental Evaluation

We experimentally evaluate the scalability of our regret-minimization algorithm for computing an extensive-form correlated equilibrium. In particular, we implement a regret minimizer for the space of correlation plans by computing the structural decomposition of $\Xi$ into a chain of scaled extensions (Section 4.3) and repeatedly applying the construction of Section 3.2. This regret minimizer is then used on the saddle-point formulation of an EFCE (Section 2.1) as explained in Section 2.3, with two modifications that are standard in the literature on regret minimization algorithms for game theory [Tammelin *et al.*, 2015; Burch *et al.*, 2019]: (i) alternating updates and (ii) linear averaging of the iterates[4]. We use *regret-matching-plus* [Tammelin *et al.*, 2015] to minimize the regret over the simplex domains in the structural decomposition. These variants are known to be beneficial in the case of Nash equilibrium, and we observed the same for EFCE. We compare our algorithm to two known algorithms in the literature. The first is based on linear programming [von Stengel and Forges, 2008].

The second is a very recent subgradient descent algorithm for this problem [Farina *et al.*, 2019c], which leverages a recent subgradient descent technique [Wang and Bertsekas, 2013]. All algorithms were run on a machine with 16 GB of RAM and an Intel i7 processor with 8 cores. We used the Gurobi commercial solver (while allowing it to use any number of threads) to solve the LP when evaluating the scalability of the LP-based method proposed by von Stengel and Forges [2008].

| Board size | Num turns | Ship length | $|\Sigma_1|$ | $|\Sigma_2|$ | Num. rel. seq. pairs |
|---|---|---|---|---|---|
| (3, 2) | 3 | 1 | 15k | 47k | 3.89M |
| (3, 2) | 4 | 1 | 145k | 306k | 26.4M |
| (3, 2) | 4 | 2 | 970k | 2.27M | 111M |

Table 1: Game metrics for the different instances of the Battleship game we test on.

**Game instances.** We test the scalability of our algorithm in a benchmark game for EFCE that was recently proposed by Farina *et al.* [2019b]: a parametric variant of the classical war game *Battleship*. Table 1 shows some statistics about the three game instances that we use, including the number of relevant sequence pairs in the game (Definition 2). 'Board size' refers to the size of the Battleship playfield; each player has a field of that size in which to place his ship. 'Num turns' refers to the maximum number of shots that each player can take (in turns). 'Ship length' is the length of the one ship that each player has. Despite the seemingly small board sizes and the presence of only one ship per player, the game trees for these instances are quite large, with each player having tens of thousands to millions of sequences.

**Scalability of the Linear Programming Approach [von Stengel and Forges, 2008].** Only the small instance could be solved by Gurobi, Figure 3 (left). (Out of the LP algorithms provided by

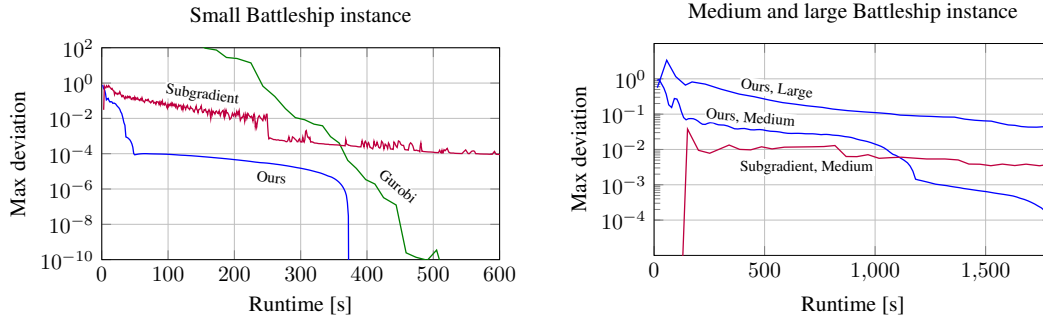

Figure 3: Experimental results. The y-axis shows the maximum utility increase upon deviation.

Gurobi, the barrier method was faster than the primal- and dual-simplex methods.) On the medium and large instance, Gurobi was killed by the system for trying to allocate too much memory. Farina *et al.* [2019c] report that the large instance needs more than 500GB of memory in order for Gurobi to run. The Gurobi run time shown in Figure 3 does not include the time needed to construct and destruct the Gurobi LP objects, which is negligible.

**Scalability of the Very Recent Subgradient Technique [Farina *et al.*, 2019c].** The very recent subgradient descent algorithm for this problem was able to solve the small and medium instances if the algorithm's step size was tuned well. An advantage of our technique is that it has no parameters to tune. Another issue is that the iterates $\Xi$ of the subgradient algorithm are not feasible while ours are. Furthermore, on the large instance, the subgradient technique was already essentially unusable because each iteration took over an hour (mainly due to computing the projection).

Figure 3 shows the experimental performance of the subgradient descent algorithm. We used a step size of $10^{-3}$ in the small instance and of $10^{-6}$ in the medium instance. Since the iterates produced by the subgradient technique are not feasible, extra care has to be taken when comparing the performance of the subgradient method to that of our approach or Gurobi. Figure 5 in Appendix G in the full paper reports the infeasibility of the iterates produced by the subgradient technique over time.

**Scalability of Our Approach.** We implemented the structural decomposition algorithm of Section 4.3. Our parallel implementation using 8 threads has a runtime of 2 seconds on the small instance, 6 seconds on the medium instance, and 40 seconds on the large instance (each result was averaged over 10 runs). Finally, we evaluated the performance of the regret minimizer constructed according to Section 3.2; the results are in Figure 3 (left) for the small instance and Figure 3 (right) for the medium and large instance. The plots do not include the time needed to construct and destruct the regret minimizers in memory, which again is negligible. As expected, on the small instance, the rate of convergence of our regret minimizer (a first-order method) is slower than that of the barrier method (a second-order method). However, the barrier method incurs a large overhead at the beginning, since Gurobi spends time factorizing the constraint matrix and computing a good ordering of variables for the elimination tree. The LP-based approach could not solve the medium or large instance, while ours could. Even on the largest instance, no more than 2GB of memory was reserved by our algorithm.

## 6  Conclusions

We introduced the first efficient regret minimization algorithm for finding an extensive-form correlated equilibrium in large two-player general-sum games with no chance moves. This is more challenging than designing an algorithm for Nash equilibrium because the constraints that define the space of correlation plans lack the hierarchical structure of sequential strategy spaces and might even form cycles. We showed that some of the constraints are redundant and can be excluded from consideration, and presented an efficient algorithm that generates the space of extensive-form correlation plans incrementally from the remaining constraints. We achieved this decomposition via a special convexity-preserving operation that we coined *scaled extension*. We showed that a regret minimizer can be designed for a scaled extension of any two convex sets, and that from the decomposition we then obtain a global regret minimizer. Our algorithm produces feasible iterates. Experiments showed that it significantly outperforms prior approaches—the LP-based approach and a very recent subgradient descent algorithm—and for larger problems it is the only viable option.

## Acknowledgments

This material is based on work supported by the National Science Foundation under grants IIS-1718457, IIS-1617590, and CCF-1733556, and the ARO under award W911NF-17-1-0082. Gabriele Farina is supported by a Facebook fellowship. Co-authors Ling and Fang are supported in part by a research grant from Lockheed Martin.

## Footnotes

[2] A feasible EFCE can be found in theoretical polynomial time [Huang and von Stengel, 2008; Huang, 2011] using the *ellipsoid-against-hope* algorithm [Papadimitriou and Roughgarden, 2008; Jiang and Leyton-Brown, 2015]. Unfortunately, that algorithm is known to not scale beyond small games.

[3]Linear-time regret minimizers for simplexes include regret-matching [Hart and Mas-Colell, 2000], regret-matching-plus [Tammelin *et al.*, 2015], mirror-descent and follow-the-regularized-leader (e.g, Hazan [2016]).

[4]The linear average of $n$ vectors $\boldsymbol{\xi}_1, \ldots, \boldsymbol{\xi}_n$ is $(\sum_{t=1}^{n} t \cdot \boldsymbol{\xi}_t)/(\sum_{t=1}^{n} t) = 2(\sum_{t=1}^{n} t \cdot \boldsymbol{\xi}_t)/(n(n+1))$.

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
