[Reviews · NeurIPS 2019]

Reviewer 1



The paper presented a new approach for working with strategies in sequential decision making problems. The new algorithm using the approach advances the state of the art for that class of problems, although the practical applications might be limited by the requirement of no chance events. The writing was fairly clear, although this paper does seem a bit cramped: there were a number of things I wish could have been expanded. That said, I don't think there is room without cutting or condensing something else, and the results stand well enough on their own that the paper should still be published as is. (Consider an expanded journal version?)

Reviewer 2



This is a well written paper with a original and significant contribution. Paper is well referenced and written very clearly. I very much appreciate that authors include the small running examples before the full construction. This makes it much easier to comprehend the ideas - and authors generally do a really good job trying to get the idea through rather than just jumping to formal constructions. The only weaker part of the paper is the evaluation section, which feels a little rushed and cramped.

Reviewer 3



**Originality** The paper presents a novel perspective on counterfactual regret minimization (CFR) in sequential problems which leads to new proof if the CFR theorem for sequence form plans and allows generalization of the result to correlated plans. As a result, it presents first CFR-based algorithm for computing EFCE. I consider the contribution in the paper to be novel. **Significance** The scaled expansion operator can potentially lead to further generalizations of CFR-based solution techniques for other problems. A scalable algorithm form computing EFCE may lead to new applications of computational game theory in the real world. Therefore, I consider the contribution to be significant. **Quality** The part of the paper that introduces scaled expansion, its properties and use to create correlation plans seems to be correct. However, the paper lacks a lot of details on how exactly do these contributions translate to a complete algorithm. What loss is optimized in each local regret minimizer? How is average strategy computed? Why does the algorithm actually converge to EFCE? I assume the answers to some of these questions are in Farina et al. (2019c) and many can be guessed with sufficient background knowledge, but the paper seems quite incomplete without this discussion. **Clarity** The paper deals with complicated topic, but it still reasonable easy to read. I think it is doing a very good job within the limited space. However, it should definitely include more (at least high level) discussion of how to turn the results presented in the paper into a complete algorithm and why the complete algorithm actually converges to EFCE. The description of the experimental domain should include more details about reward structure to see that it is sufficiently non-zero-sum. ** After Rebuttal ** Thank you for your response. Even with the response, I still do not know how to implement the algorithm so it is not likely that you can make the paper more self-contained in the given space. With the averaging, rather than explaining what is linear averaging, it would be more useful (in the final paper) to comment on why it is OK to simply average the correlation plans, since it would not work with behavior strategies, which are more common in the context of CFR.

[Author Response · NeurIPS 2019]

**Response to Reviewer 1** Thanks for all the remarks and the useful suggestions. We'll address all of them in the final version of the paper.

- **Re "The number of constraints is bounded by** $1 + |I_1||\Sigma_2| + |\Sigma_1||I_2|$**?"** Yes, exactly. We will add a remark about this after we define $\Xi$ in Definition 3.

- **Re "Scalability of the Very Recent Subgradient Technique"** Good point. We originally chose to not give much space to the technique of Farina et al., since it does not guarantee feasible iterates and therefore the comparison with our algorithm and the linear programming approach is not apples-to-apples. You can see the performance of the subgradient technique for the Small instance in Figure 1; the plot does not account for the computation of the sparse factorization needed by the algorithm, which took $\approx$500ms. The algorithm was run on the same machine and under the same conditions described in the experimental section of our paper. The stepsize used was 0.001. We will add a few sentences to the final version and include this plot in the appendix.

- **Re "Is there some explanation for how the max deviation drops so much on the "ours" line, when the text is suggesting it is doing a** $t \cdot x^t$ **weighted average?"** Good question. We tried to look into it after we ran the experiments, but we don't have any definite answer. Our best guess is that the polytope of EFCE has interior, and that the gap drops suddenly because the average of the iterates entered the polytope of EFCE. The choice of using linear averaging was popularized by the original CFR$^+$ work, and is now standard in the computational game theory literature.

Figure 1: Performance of the subgradient technique of Farina et al. (2019c). The infeasibility vector of an iterate $\boldsymbol{\xi}$ is defined as the vector of absolute differences between the left-hand and right-hand sides of all the constraints that define $\Xi$ (Definition 3).

**Response to Reviewer #2** Thanks for the nice and thorough review, and for catching that typo on line 232! We will add a reference to the suggested JAIR article.

- **Question 1** In the specific case of the sequence-form strategy polytope, the top-down (i.e., using scaled extension) and the bottom-up (i.e., using convex hulls and Cartesian products) constructions lead to the *same* regret minimizer. So, for that application, there is no difference between the two approaches. We did not state that in the reviewed version in order to keep the focus on EFCEs. (Based on your comment, we will state the equivalence for the sequence-form case in the camera ready.)

- **Question 2** Broadly speaking, the problem of the bottom-up approach is that it cannot capture the affine structure (Line 302 in the algorithm, lines 235-241 in the example).

- **Question 5** The plots currently don't include the time it took to construct/deconstruct the regret minimizers, as much as they don't include the time it took to construct/deconstruct the Gurobi LP objects. The difference in the plots from including those times would be nearly invisible. We will make sure to point this out in the final version, thanks!

**Response to Reviewer #3** Thanks for your review. Because of the tight space constraints, in the paper we were forced to take the saddle-point formulation of the EFCE problem as a given, and only focus on the really novel part, that is show how to construct an efficient regret minimizer for the $\Xi$ polytope. In order to make the paper more self-contained, we will add an appendix in which we restate the saddle-point formulation of EFCE, and give a description of the full algorithm with pseudocode.

- **Re "Why does the algorithm actually converge to EFCE?"** As noted in Section 2.2, the problem of computing an EFCE is a bilinear saddle-point problem (BSPP), that is a problem of the form $\min_{\boldsymbol{x} \in \mathcal{X}} \max_{\boldsymbol{y} \in \mathcal{Y}} \boldsymbol{x}^\top \boldsymbol{A} \boldsymbol{y}$ where $\mathcal{X} = \Xi$ is the convex polytope defined in Definition 3, $\mathcal{Y}$ is a convex and compact set (more details in Farina et al. 2019c), and $\boldsymbol{A}$ is a real matrix. It is known that *any* BSPP can be solved using regret minimization (Section 2.4). In particular, given access to two regret minimizers that output decisions on $\Xi$ and $\mathcal{Y}$, respectively, we can compute a saddle point by letting the two regret minimizers face each other. In this paper we construct an efficient regret minimizer that can output decisions on the set $\Xi$.

- **Re "What loss is optimized in each local regret minimizer?"** The loss vector that is received by the overall regret minimizer for $\Xi$ is as in Farina et al. (2019c); as mentioned above, we will dedicate a new appendix to recalling the details of how that vector is computed. From there, the loss is split into smaller vectors that are input to each local regret minimizer according to the OBSERVELOSS procedure (Algorithm 1 in our paper).

- **Re "How is average strategy computed?"** The average $\boldsymbol{\xi}$ strategy is computed using *linear averaging* (Line 324), a standard technique in computational game theory literature. The linear averaging of $n$ vectors $\boldsymbol{\xi}_1, \ldots, \boldsymbol{\xi}_n$ is defined as the weighted average $(\sum_{t=1}^{n} t \cdot \boldsymbol{\xi}_t)/(\sum_{t=1}^{n} t) = 2(\sum_{t=1}^{n} t \cdot \boldsymbol{\xi}_t)/(n(n+1))$. We'll remind the reader of this definition in the final version.

[Meta-Review · NeurIPS 2019]

A well written paper with clear contributions. It would even benefit from having an "extended" (or journal) version. The next step will definitely to add "chance moves" or uncertainties !